# Observation of the Antimicrobial Activities of Two Actinomycetes in the Harvester Ant *Messor orientalis*

**DOI:** 10.3390/insects13080691

**Published:** 2022-07-31

**Authors:** Yiyang Wu, Yaxuan Liu, Jinyong Yu, Yijuan Xu, Siqi Chen

**Affiliations:** 1Guangdong Laboratory for Lingnan Modern Agriculture, Red Imported Fire Ant Research Center, South China Agricultural University, Guangzhou 510642, China; yiyang_hongling@163.com; 2Sendelta International Academy, Shenzhen 518000, China; 3Department of Material Science and Engineering, College of Engineering, Carnegie Mellon University, Pittsburgh, PA 15213-2683, USA; yaxuanl2@andrew.cmu.edu; 4College of Agronomy and Biotechnology, Hebei Normal University of Science & Technology, Qinhuangdao 066600, China; yujing_75211@163.com

**Keywords:** *Messor orientalis*, harvester ants, actinomycetes, plant pathogens, fungicide

## Abstract

**Simple Summary:**

Observations in the animal room have shown that the seeds stored by harvester ants, although in a damp environment, are less likely to mold. It was hypothesized that harvester ants may use actinomycetes to protect their seed stores, given that leafcutter ants use actinomycetes as producers of defensive substances. Two actinomycetes were isolated from the harvester ant *Messor orientalis*. The fermentation broth of the actinomycetes showed significant inhibitory effects on the three indicator fungi. Coculture experiments supported the observed inhibitory effects. The antifungal activities of actinomycetes in harvester ants were revealed. This research provides a significant theoretical reference for the abovementioned hypothesis and for the potential agricultural applications of these actinomycetes for multiple crops.

**Abstract:**

Observations have shown that seeds collected by harvester ants are less likely to mold. Based on evolutionary analysis and other research, it was hypothesized that harvester ants could apply actinomycetes to protect seeds, similar to the protection of mutualistic fungi by leafcutter ants. Two actinomycetes were successfully isolated from the harvester ant *Messor orientalis.* The taxonomic status of the actinomycetes was determined by 16S rRNA sequence analysis and biochemical experimental observations. Their inhibitory effects on plant pathogens were measured. One of the bacteria was identified as *Brachybacterium phenoliresistens* and denoted as *B. phenoliresistens MO*. The other belonged to the genus *Microbacterium*. It was named *Microbacterium* sp. Growth rate determination and coculture experiments were performed to explore the inhibitory effect of actinomycetes on indicator plant pathogens. The inhibition rates of the actinomycetes toward *Peronophythora litchii* and *Rhizoctonia solani* were 100% in media containing 30% or more fermentation broth, and they also showed an inhibitory effect on *Colletotrichum siamense*. The coculture experiment supported this result by showing that the growth of *P. litchii* and *R. solani* was inhibited in the presence of actinomycetes. Therefore, the results of this study show the agricultural application potential of these bacteria and may provide a reference for research on the symbiosis of harvester ants with actinomycetes.

## 1. Introduction

Symbiotic microbes exist in a vast majority of insects [1], with actinomycetes accounting for a large proportion of the microbes found in insects [2]. Actinomycetes can help insects adapt to their habitats and resist natural enemies, and they even play a dominant role in the food digestion process [3,4,5]. Various metabolites of symbiotic actinomycetes derived from insects have shown bacteriostatic activity. Lu et al. isolated a wide range of metabolites from the actinomycete *Streptomyces violaceoruber BYC-01*, which showed inhibitory effects on fungi obtained from termite nests. A single compound, fogacin, was extracted from the fermentation broth of this strain via distillation and ethyl acetate extraction [6,7,8]. Insect-derived *Streptomyces* species exhibit high inhibitory activity [9]. *Streptomyces* species symbiotically associated with *Dendroctonus frontalis* produce the secondary metabolites frontalamides and mycangimycin [9,10,11]. Mycangimycin inhibits the beetle’s antagonistic fungus *Ophiostoma minus*, while frontalamides have general antifungal activity [9,10,11]. Sceliphrolactam, an antifungal compound isolated from *Streptomyces,* was found to be associated with a mud dauber wasp [9,12]. These results are promising, and there is a wide variety of insects in the world, so research on insect-associated microorganisms has high application potential, and several abundant sources of active metabolites have yet to be explored. Ants play an indispensable regulatory role in the ecosystem [13]. Studies have shown that leafcutter ants place parasitic fungal spores near symbiotic actinomycetes, *e.g.*, *Nocardiopsis*, until these spores lose their infectivity [14]. The coexistence of *Streptomyces* sp. with strong bacteriostatic properties has been reported in invasive fire ants (*Solenopsis invicta Buren*) [15]. These studies showed that actinomycetes in insects may provide abundant resources for the development of antimicrobial agents. These symbioses are best exemplified in fungus-growing ants [5,8,16], carpenter ant [17,18,19], solitary digger wasps [12], and southern pine beetles [11].

A study by Tarsa et al. revealed a negative correlation between the occurrence of seed-collecting ants and that of plant pathogens [20]. Harvester ants, i.e., *Messor Forel*, collect and store plant seeds in their nests, which may affect microbial composition [21]. Observations in the animal room have shown that the seeds stored by harvester ants are less likely to mold, but the underlying mechanism remains unclear. Based on analyses from previous studies, actinomycetes in harvester ants may play an important role in this phenomenon.

According to Kang, 10–15% of agricultural production in the world is lost due to improper storage and diseases, among other reasons [22], and 70–80% of the total loss is attributed to plant pathogenic fungi. Thus, to mitigate the loss of grain to plant pathogenic fungi, effective fungicides must be developed. The recent development of fungicides involves the use of plant extracts and the isolation of new compounds from microbial metabolites [23]. Actinomycetes are highly applicable in biological control because their metabolites possess strong bacteriostatic properties [24,25].

In this study, actinomycetes were isolated from *Messor orientalis*, and two actinomycete species belonging to the genera *Microbacterium* and *Brachybacterium* were analyzed. The antimicrobial activity of symbiotic actinomycetes against fungi such as *Colletotrichum siamense* was assayed. The results obtained in this work may provide a scientific reference for the development of new fungicides and aid future research on seed protection mechanisms.

## 2. Materials and Methods

### 2.1. Collection of Ant Samples and Indicator Fungi

The ant samples were collected by local collectors in Wujiaqu, Xinjiang, China, in 2020. The ants were kept in test tubes in a storage box, which was placed in an animal culture room at a constant temperature of 27 °C. The ants were morphologically identified as *M. orientalis*.

Three plant pathogens, i.e., *C. siamense* (collection code: CSGD18001), *Peronophythora litchii* (collection code: PLGD18001), and *Rhizoctonia solani* (collection code: RSGD18001), were kindly provided by the Plant Fungi Laboratory, South China Agricultural University.

### 2.2. Isolation of Actinomycetes

The collected ants were provided with water and *Phalarit canariensis* seeds as food every day. To avoid the influence of disturbance during collection on the community of symbiotic actinomycetes, we let the ant colony stabilize indoors for two weeks before the isolation of actinomycetes. Twenty-five harvester ants from 4 different colonies (6–7 ants per colony) were washed with sterilized water and 70% alcohol. Then, the ants in each group were rinsed twice with sterilized water. The ants were ground with 1 mL water to obtain a liquid ground-ant sample.

Dilutions of the liquid ground-ant sample (10^−2^, 10^−3^, and 10^−4^) were plated on sterilized Gauze’s No. 1 agar medium (every 150 mL was amended with 25 mg of cycloheximide and 8 mg of nalidixic acid). Each dilution was repeated 3 times. Then, the samples were sealed and kept at 28 °C for 7 days. The actinomycete colonies obtained were then inoculated on sterilized Gauze’s No. 1 medium and cultured at 28 °C for 7 days.

### 2.3. 16S rRNA Gene Sequencing and Biochemical Identification

A single colony of each actinomycete was used to extract the pure genomic DNA using the Bacterial DNA Extraction Kit (Tiangen Biotech (Beijing) Co., Ltd.) according to the manufacturer’s instructions. Primers 27F (5′-AGAGTTTGATCCTGGCTCAG-3′) and 1492R (5′-GGTTACCTTGTTACGACT-3′) were used as universal primers for bacterial 16S rRNA amplification. The total PCR volume was 25 μL, including 12.5 μL of 2×Taq PCR Mastermix (Tiangen Biotech (Beijing) Co., Ltd., Beijing, China), 1 μL of each primer (10 µM), 1.5 μL of DNA template, and 9 μL of ddH_2_O. The PCR protocol is shown in Table 1.

The PCR products were sent to RuiBiotech for Sanger sequencing. Reads greater than 1400 bp in length were used for the database analysis. The sequencing results were compared using NCBI nucleotide BLAST (https://blast.ncbi.nlm.nih.gov/Blast.cgi, accessed on 20 July 2022). Then, 16S rRNA sequences with the highest similarity to those of the isolated strains and typical strains in the same genus were obtained for phylogenetic analysis. The phylogenetic trees evaluated with 1000 bootstrap replications were inferred using the maximum likelihood method based on the Tamura 3-parameter model in MEGA7 (GenBank accession number: OM665406; OM665407).

Inositol, maltose, dextrose, rhamnose, sucrose, Neisser-fructose, hydrogen sulfide production, xylose, mannitol, and raffinose identification tubes purchased from Huancai Microbial Technology were used to test the substrate utilization of the isolated strains. A total of 50 μL of activated broth was added to the identification tubes and then cultured at 37 °C for 24 h. Substrate utilization data for other species in similar genera were obtained for comparison with the data for the isolated strains. Gram staining of the colonies was conducted and then observed microscopically.

### 2.4. Bioassay of the Fungal Inhibition Effect

A single colony of actinomycetes was activated in liquid BHI medium (37 °C ± 1 °C, 160 r/min) for 24 h. Then, 1 mL of the activated broth was added to a 250 mL flask that contained 100 mL of soybean powder fermentation broth. The mixture was cultured in a shaker (28 °C ± 1 °C, 160 r/min) for 7 days. The fermentation broth was centrifuged (4 °C, 8000 r/min, 20 min) and filtered using a 0.22 µm filter membrane to obtain the aseptic filtrate. Various amounts of filtrate (1 mL, 1.5 mL, and 2 mL) were added to sterilized PDA medium (4 mL, 3.5 mL, and 3 mL) on a plate with a diameter of 5.5 cm. Then, the medium was allowed to cool to 55 °C. The control PDA plate was supplemented with various amounts of sterile water (1 mL, 1.5 mL, and 2 mL). Plant pathogenic fungal plugs with a diameter of 0.5 cm were placed on PDA medium. For each actinomycete sample, the experiment was repeated three times on media containing each fermentation broth filtrate dilution for each fungus. The average diameters of the fungal colonies in the experimental and control groups were recorded. The bacteriostatic rate of the actinomycete against the three-indicator plant pathogenic fungi was calculated according to the following equation:$$\mathrm{Inhibition}\;\mathrm{rate}\%=\frac{\left(\mathrm{ADc}-D\right)-\left(\mathrm{ADt}-D\right)}{\left(\mathrm{ADc}-D\right)\;}\times 100\%$$
where ADc represents the average colony diameter in the control group, ADt represents the average colony diameter in the treatment group, and D represents the diameter of the fungal plugs.

The inhibitory effect on phytopathogens was shown directly by the confrontation culture method. Circular plugs of fungi (diameter = 0.5 cm) were placed in one-quarter of the plates after the actinobacterial plugs were placed in the other quarter of the plates for 7 days at 28 °C.

The Kruskal–Wallis (KW) nonparametric analysis of variance was used to compare the different treatments. The Mann–Whitney U test for multiple comparisons among the different groups was used if the results of the Kruskal–Wallis test showed significant differences at a significance level of 0.05.

## 3. Results

### 3.1. Identification of Actinomycetes

Two strains of bacteria, A and B, were identified in *M. orientalis* based on morphological observation and 16S rRNA sequencing. Figure 1 shows the results obtained from the 1.5% agarose gel electrophoresis of the PCR products. Clear bands corresponding to a length of 1500 bp were observed.

The 16S rRNA sequences (GenBank accession number: OM665406; OM665407) of the two bacteria were compared using NCBI nucleotide BLAST. Based on the results, these two bacteria were identified as actinomycetes belonging to the genera *Microbacterium* and *Brachybacterium*. The similarity between strain A and *Brachybacterium phenoliresistens* was 99.86%. The similarity between strain B and *Microbacterium barkeri* was 99.34%. The physiological and biochemical characteristics of strains A and B are shown in Table 2 and Table 3, respectively. The results indicated that these two actinomycetes were successfully identified. However, it is notable that the rhamnose utilization capacity of strain B differed from that of *M. barkeri*.

Figure 2 shows the phylogenetic tree that was constructed based on 20 known strains of *Brachybacterium* and strain A (shown as *Brachybacterium* sp.). As shown in the phylogenetic tree, *B. phenoliresistens* and strain A were closely related, with high repeatability. Figure 3 shows the phylogenetic tree that was constructed based on 11 strains with the highest similarity to strain B (shown as *Microbacterium* sp.), according to BLAST. The findings showed that strain B and *M. barkeri* were closely related.

Based on the analysis, strain A was preliminarily considered to be a *Brachybacterium* strain, probably a strain of *B. phenoliresistens*. Hence, strain A is denoted *Brachybacterium* sp. *MO*. Strain B belongs to the *Microbacterium* genus. However, strain B has yet to be identified by multiphase classification and identification, so it is denoted as *Microbacterium* sp.

### 3.2. Study of Fungal Inhibitory Activity

#### 3.2.1. Fungal Inhibitory Activity of the Fermentation Broth

The inhibitory effects of *B. phenoliresistens MO* and *Microbacterium* sp. on three types of plant pathogenic fungi with different fermentation broth concentrations are shown in Figure 4 and Figure 5, respectively. Significant inhibitory effects were exhibited by these two actinomycete fermentation broths on the three plant pathogens. As the fermentation broth filtrate concentration increased, the inhibitory effect of the bacteria on plant pathogenic fungi increased. As shown in Figure 4 and Figure 5, the inhibitory rates exhibited by these two actinomycetes on *R. solani* and *P. litchii* were both 100%. However, the inhibitory effects of both actinomycetes on *C. siamense* were weaker. To illustrate these results visually, digital photographs of the pathogenic fungi on the fermentation broth filtrate plate are presented in Appendix A. The growth of these plant pathogenic fungi on the fermentation filtrate plate was inhibited.

#### 3.2.2. Inhibitory Activity of Actinomycete Colonies

On the bacterial plate, the growth of the pathogen was almost unaffected by live *B. phenoliresistens MO* and *Microbacterium* sp., which was consistent with the results obtained from the fermentation broth inhibition experiment. *P. litchii* growth was significantly inhibited by strains A and B. Mycelial growth around live strain B was relatively unaffected, while germination could not occur around live strain A, as shown in Figure 6.

## 4. Conclusions

Two actinomycete strains were isolated from the harvester ants. Based on the 16S rRNA and substrate utilization analysis, strain A belongs to the *Brachybacterium* genus, and it is very likely that *B. phenoliresistens*. Hence, strain A is denoted *B. phenoliresistens MO*. Strain B belongs to the *Microbacterium* genus. Because there were differences in the substrate utilization results, strain B must be further classified. It is denoted here as *Microbacterium* sp.

The inhibitory effects of the 2 actinomycetes on plant pathogenic fungi were assayed. When the actinomycete fermentation broth concentration was higher than 30%, the inhibition rates on the indicator fungi, i.e., *P. litchii* and *R. solani,* were significantly high, wherein a 100% inhibition rate was recorded. The inhibitory effects of these two actinomycetes on *C. siamense* were also notable, indicating prospects for agricultural applications. Through coculturing live bacteria, we obtained results consistent with a previous conclusion, showing that the two strains exhibited strong inhibitory effects on *P. litchii* and *R. solani*.

## 5. Discussion

Rice sheath blight caused by *R. solani* is one of the three major rice diseases and is the major disease in rice-producing areas in Asia. As a result, the damage caused by *R. solani* has led to significant agricultural losses in China each year [26]. In addition, infestation by *P. litchii* in litchi during storage and transport can result in great losses each year [27]. In this work, two strains of actinomycetes were used on these two types of plant pathogenic fungi at certain concentrations, and an inhibition rate of 100% was observed; therefore, the inhibitory effect of the live bacterial cultures on the pathogenic fungi was confirmed. However, the exact mechanism remains unclear. This result is possibly related to the bacterial fermentation products. In general, the inhibitory rates exhibited by the actinomycetes on *P. litchii* were higher than those of the other two pathogenic fungi, which suggests that *P. litchi* is more sensitive to actinomycetes [15]. The 100% pathogenic fungus inhibition rate may indicate the potential agricultural applications of these actinomycetes for multiple crops. Multiple methods—HPLC, metabolomic profiling, and gas chromatography—can be used to study the chemical characteristics, including toxicity, light degradability, and stability, of antifungal metabolites to determine whether actinomycetes can be used as fungicides against *R. solani* and *P. litchii*. As such, this work indicates a potential means of reducing the economic loss caused by damage to rice and litchi by *R. solani* and *P. litchii*.

*Microbacterium* is an abundant component of bacterial communities in the soil, insects, and leaf material of plants [8]. In previous studies, *M. testaceum* KU313 isolated from stored rice grains was antagonistic to *Aspergillus flavus* and *Penicillium* spp. [28]. *Microbacterium* sp. LGMB471 isolated from the medicinal plant *Vochysia divergens* inhibited the development of *Phyllosticta citricarpa* [29]. *Microbacterium* sp. isolated from tomato plants inhibited the growth of *Alternaria alternata*, *Corynespora cassiicola,* and *Stemphylium lycopersici* [30]. Among these, antifungal compounds, 5-methyl-2-phenyl-1H-indole produced by strain KU143, 7-O-β-D-glucosyl-genistein and 7-O-β-D-glucosyl-daidzein produced by strain LGMB471, were also discovered [28,29]. *Brachybacterium* exists in various environments, such as soil (poultry deep litter, contaminated sand), roots, fermented food, and animals [31]. It was reported that *B. paraconglomeratum* YEBPT2 isolated from banana contributed to antifungal activity against *Fusarium oxysporum* f.sp. *cubense* (Foc), and nine bioactive metabolites were identified as diethyl hydrazine, carbonic acid, nitrosopyrrolidine, 4H-pyran, valeric acid, butanoic acid, trioxsalen, deoxy-d-mannoic acid, and amino caprylic acid [32].

Ants are one of the most successful terrestrial species [33]. Many researchers devoted to screening the high biotechnological potential of ant-associated microorganisms, as well as the significant ecological impact of microbial secondary metabolites [5,8,9,13,14,15,16,17,18,34,35,36]. Nonetheless, most studies have mainly focused on ant–fungus–actinomycete tripartite mutualism evolved by leaf-cutting ants, which use antifungal microbial secondary metabolites produced by actinobacteria (*Streptomyces* spp., *Nocardia* spp., *Pseudonocardia* spp., *Amycolatopsis* spp., etc.) to control pathogens in their fungal gardens [5,9,16,35,36,37,38,39]. The rich diversity of antimicrobial secondary metabolites plays a driving role in shaping the ecosystems of leaf-cutting ants. Only by revealing the chemical nature of antibiotics can we begin to fully understand the complex interactions between multi-organismic partners. In this work, *B. phenoliresistens MO*. and *Microbacterium* sp. exhibited pronounced antifungal properties; however, the possible symbiotic relationship between harvester ants and actinomycetes remains unclear. The genera *Microbacterium* isolated from gardens and starter cultures of *Atta* could play disease-suppressing or other unknown roles [8], while no specific function was shown for *Brachybacterium* isolated from the abdomen of *Leucocoprinus gongylophorus* [40]. To better understand the ecological role of microorganisms associated with *Messor orientalis*, it is crucial to analyze the chemical composition and evaluate the biological activity of their metabolites. In further work in this area, we wish to identify more species of functional actinomycetes from *Messor orientalis* by using diverse isolation methods and media. In addition, we will further examine the relationship between harvester ants and actinomycetes to understand whether the ants and microbes have a mutually beneficial relationship.

## Figures and Tables

**Figure 1 insects-13-00691-f001:**
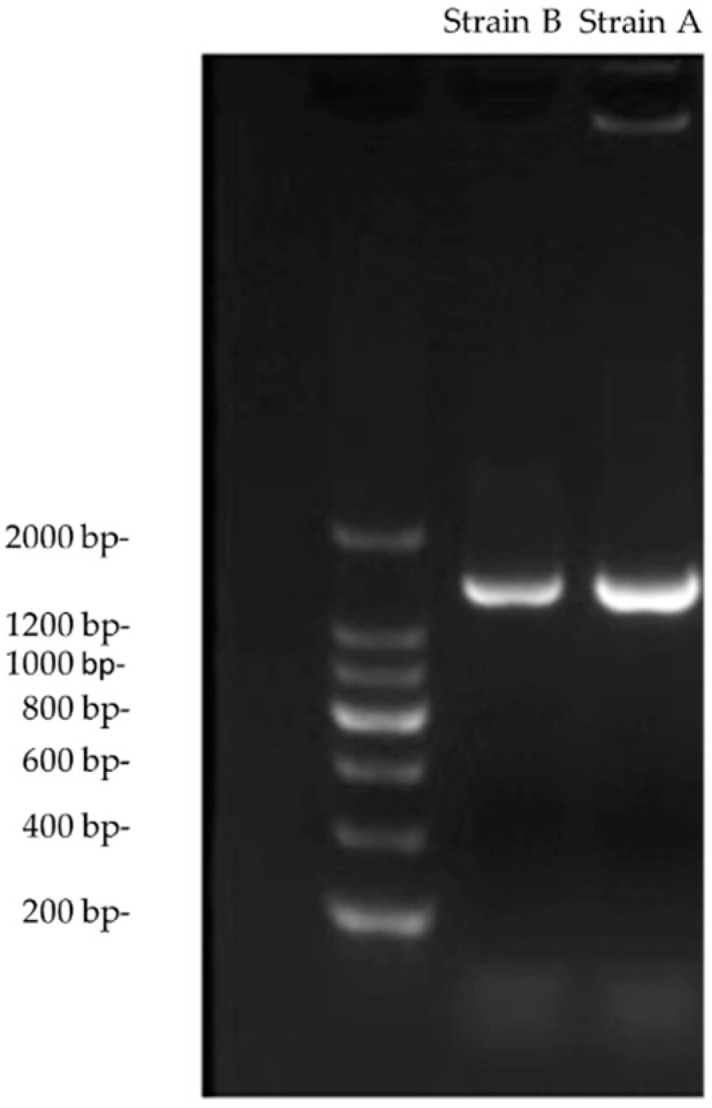
PCR amplification and gel electrophoresis of strains A and B.

**Figure 2 insects-13-00691-f002:**
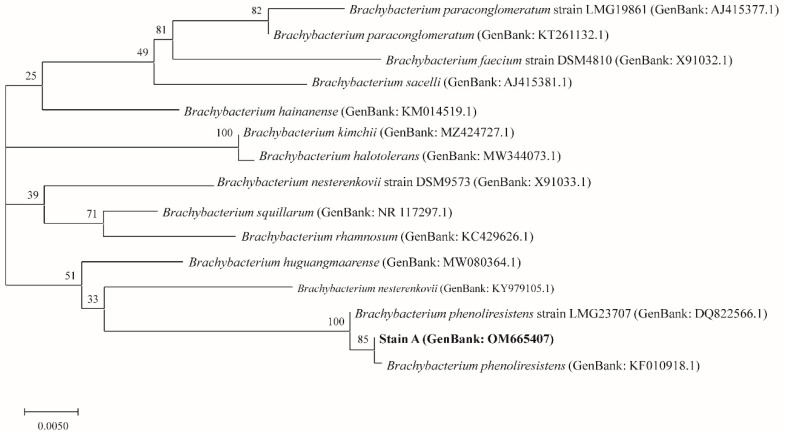
Molecular phylogenetic analysis of strain A (*Brachybacterium* sp.) using the maximum likelihood method.

**Figure 3 insects-13-00691-f003:**
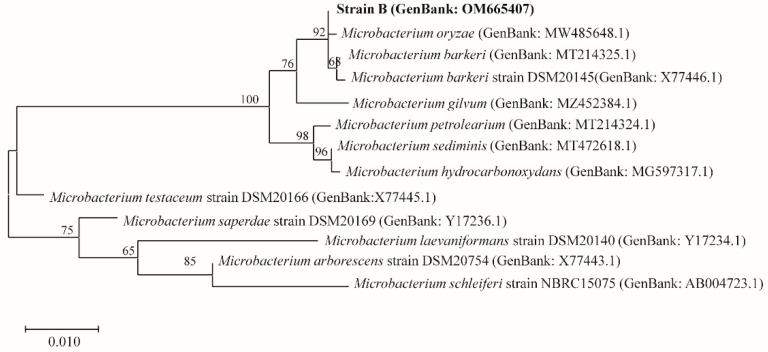
Molecular phylogenetic analysis of strain B (*Microbacterium* sp.) using the maximum likelihood method.

**Figure 4 insects-13-00691-f004:**
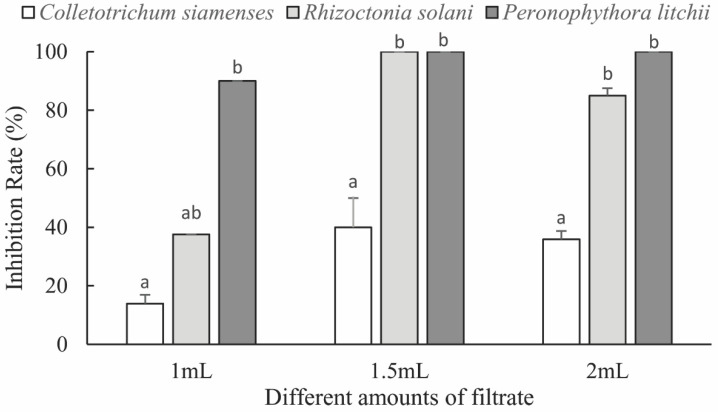
Inhibitory effect of *Brachybacterium phenoliresistens* against three plant pathogens with different amounts of filtrate. For each amount of filtrate, bars with the same letter are not significantly different (*p* > 0.05, Mann–Whitney U test).

**Figure 5 insects-13-00691-f005:**
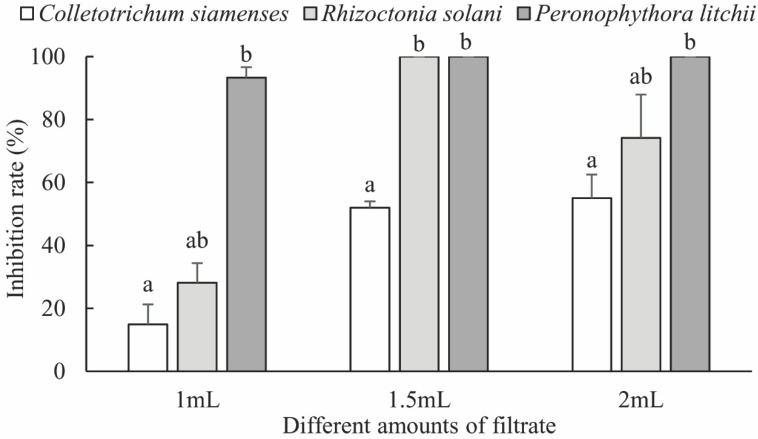
Inhibitory effect of *Microbacterium* sp. against three plant pathogens with different amounts of filtrate. For each amount of filtrate, bars with the same letter are not significantly different (*p* > 0.05, Mann–Whitney U test).

**Figure 6 insects-13-00691-f006:**
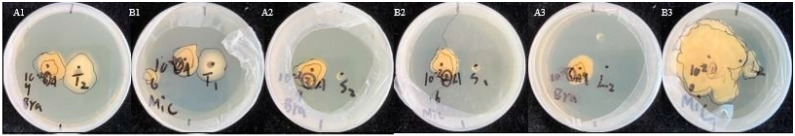
Cogrowth of actinomycetes and fungi. (**A**) *Brachybacterium phenoliresistens* MO. (**B**) *Microbacterium* sp. (**1**) *Colletotrichum siamense*, (**2**) *Rhizoctonia solani*, and (**3**) *P**eronophthora litchii*.

**Table 1 insects-13-00691-t001:** Actinomycete 16S rRNA amplification PCR protocol.

Step	Reaction Temperature (°C)	Reaction Time (min)
Initialization	94	15
Denaturing	94	0.5 ^a^
Annealing	55	0.5 ^a^
Elongation	72	1 ^a^
Stop	72	10

^a^ Step was repeated 30 times.

**Table 2 insects-13-00691-t002:** Physiological and biochemical characteristics of strain A. Strains: 1, *B. phenoliresistens* LMG 23707^T^; 2, *B. saceli* DSM 14566^T^; 3, *B. alimentarium* CCM 4520^T^; 4, *B. freconis* DSM 14564^T^; 5, *B. paraconglomeratum* DSM 46361^T^; 6, *B. faecium* CCM 4372^T^. Abbreviations: +, positive; (+), weakly positive; −, negative; ND, not determined.

Characteristic	Strain A	1	2	3	4	5	6
H_2_S production	−	−	−	+	+	+	−
Acid production from:							
D-fructose	+	ND	+	−	+	+	−
Maltose	+	+	+	−	+	+	(+)
D-mannose	+	ND	(+)	(+)	+	+	−
L-rhamnose	+	+	(+)	(+)	+	+	−
Sucrose	+	+	−	+	(+)	−	−
D-xylose	+	+	−	−	−	+	−
Galactose	+	ND	+	+	+	+	+

**Table 3 insects-13-00691-t003:** Physiological and biochemical characteristics of strain B. Strains: 1, *M. barkeri* DSM 20145^T^; 2, *M. chocolatum* IFO 3758^T^; 3, *M. hominis* IFO 15708^T^; 4, *M. thalassium* IFO 16060^T^, IFO 16061; 5, *M. halophilum* IFO 16062^T^; 6, *M. laevaniformans* IFO 15709^T^. Abbreviations: +, positive; −, negative; ND, not determined.

Characteristic	Strain B	1	2	3	4	5	6
H_2_S production	−	ND	+	+	−	−	+
Utilization of:							
Maltose	+	ND	+	+	+	+	+
D-mannose	−	ND	+	+	+	+	+
Acid production from:							
L-rhamnose	−	+	−	−	−	+	−
Sucrose	+	ND	+	+	+	−	−
D-xylose	−	ND	−	−	−	+	−
Galactose	−	+	−	+	−	−	+

## Data Availability

The data presented in this study are available on request from the corresponding author.

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
