# Peer review of "Observation of the Antimicrobial Activities of Two Actinomycetes in the Harvester Ant *Messor orientalis"

_insects, 2022, doi:10.3390/insects13080691_

Round 1
Reviewer 1 Report
I have carefully read the submitted manuscript and I think that in the present version the experimental design is adequate and the same is true for the quality of the introduction and the description of both methods and results.
The discussion has been properly focussed and the same is true for the introduction.
As a whole, I think that the manuscript should be accepted in the present version without request changes.
Author Response
We thank the reviewer's comments.
Reviewer 2 Report
The authors have significantly improved the manuscript, worked on the design, it is commendable.
However, they still need to work on the style. But there are still questions about the essence of the work - it is too simple and not informative enough.
Line 64
There is a series of works where antibiotic producers from other ant species are also identified. I recommend reading them and including them in the review.
doi:10.3390/microorganisms8121948
10.1016/j.biochi.2019.02.010
https://www.nature.com/articles/s41589-020-0578-x
Line 97
It is still not entirely clear why the authors kept the ants for two weeks after harvesting and did not carry out the isolation. How the insects were kept, whether they were fed and other factors that could affect the microbiota of insects remained unknown…
Lines 99-100
Line 102
In the previous version, the authors called this substance “actinone”, now “actinidone”. I recommend using databases and carefully checking the correctness of terms.
https://pubchem.ncbi.nlm.nih.gov/compound/Cycloheximide
Line 107
The authors ignored the remark made in the last review. I have to repeat it
“The authors did not use any methods to isolate and purify DNA for amplification, that raises doubts about the quality of the obtained results”
Figure 2 and 3.
To construct phylogenetic trees, it is necessary to use sequences of typical strains from world-famous collections of microorganisms. Strains with unknown indices may not be identified correctly.
Figure 4 and 5.
There are no explanations of what the letters a, b and ab mean.
Lines 266-284
Instead of discussing their results, the authors generously promise to do good research in the future, involving a wide range of methods. It is unclear what prevented you from doing all the research first -- and then writing an article.
In fact, there is nothing to discuss, the volume and value of the work is not great yet.
Author Response
Line 64
There is a series of works where antibiotic producers from other ant species are also identified. I recommend reading them and including them in the review.
doi:10.3390/microorganisms8121948
10.1016/j.biochi.2019.02.010
https://www.nature.com/articles/s41589-020-0578-x
Authors response: we cite these works in the revised ms. See Line 63.
Line 97
It is still not entirely clear why the authors kept the ants for two weeks after harvesting and did not carry out the isolation. How the insects were kept, whether they were fed and other factors that could affect the microbiota of insects remained unknown…
Authors response: we revised accordingly “The collected ants were provided with water and Phalarit canariensis seeds as food every day. In order to avoid the influence of disturbance during collection on the community of symbiotic actinomycetes, we let the ant colony stabilize indoors for two weeks before isolation of actinomycetes.” See Line 96-99.
Lines 99-100
Line 102
In the previous version, the authors called this substance “actinone”, now “actinidone”. I recommend using databases and carefully checking the correctness of terms.
https://pubchem.ncbi.nlm.nih.gov/compound/Cycloheximide
Authors response: we thank the reviewer’s comment. The terms have been checked online and revised. See Line 104-105.
Line 107
The authors ignored the remark made in the last review. I have to repeat it
“The authors did not use any methods to isolate and purify DNA for amplification, that raises doubts about the quality of the obtained results”
Authors response: Pure genomic DNA of actinomycetes was extracted by using the kit by Bacterial DNA Extraction Kit (Tiangen Biotech (Beijing) Co., Ltd.) according to the manufacturer’s instructions. See Line 109-111.
Figure 2 and 3.
To construct phylogenetic trees, it is necessary to use sequences of typical strains from world-famous collections of microorganisms. Strains with unknown indices may not be identified correctly.
Authors response: we thank the reviewer’s comment and have reconstruct the phylogenetic trees.
Figure 4 and 5.
There are no explanations of what the letters a, b and ab mean.
Authors response: Actually, we stated in the figure legends “For each amount of filtrate, bars with the same letter are not significantly different (p > 0.05, Mann–Whitney U test).”
Lines 266-284
Instead of discussing their results, the authors generously promise to do good research in the future, involving a wide range of methods. It is unclear what prevented you from doing all the research first -- and then writing an article.
In fact, there is nothing to discuss, the volume and value of the work is not great yet.
Authors response: we rewrite the discussion section by according to the reviewer’s comment. See the last paragraph.

Round 2
Reviewer 2 Report
The authors improve their manuscript every time, however, it still requires correction.
Fig. 2, 3
Earlier I made the remark, that
«To construct phylogenetic trees, it is necessary to use sequences of typical strains from world-famous collections of microorganisms. Strains with unknown indices may not be identified correctly»
The authors replied that they had changed the trees.
Now I see that they have removed any designation of collections and strain numbers from the trees at all. Reliable sources of typical strains are World Collections of Microorganisms such as DSMZ, ATCC, NRRL et al.
The nucleotide sequences of strains from these collections should be used to build trees.
Discussion
The "Discussion" section still does not correspond to its essence. The authors have added general arguments, which correspond to the Introduction section. Discussion implies understanding your own results.
The genera Microbacterium isolated from gardens and starter-cultures of Atta could play disease-suppressing or other unknown roles[8], while no specific function was shown for Brachybacterium isolated from the abdomen of Leucocoprinus gongylophorus[35].
Perhaps it is worth expanding the description of the ecology of close species (Brachybacterium and Microbacterium) that has begun --- where they were isolated from, what metabolites they have, and so on. Work more carefully with the literature to collect more information on these genera/species.
Also check for spaces in the text, for example, Line 130
Author Response
Please see the attachment.

This manuscript is a resubmission of an earlier submission. The following is a list of the peer review reports and author responses from that submission.
Round 1
Reviewer 1 Report
The authors have made changes and improvements to their work, but it still contains many disadvantages and negligence. (has a lot of errors and inaccuracies, is written in poor English)
Line 12
Here need to fill in E-mail and phone number of the author who is responsible for correspondence with the editorial office.
Line 18
“…as antibiotics ” it is better to replace with “… as producers of defensive substances”
Lines 14-22
Simple summary should contain more information about the possible practical benefits of this work, understandable to a wide audience.
Line 56
There are still not enough relevant articles and reviews was used as references.
It should be recommended that the authors carefully study this review
https://doi.org/10.1038/s41467-019-08438-0
Line 118
The new name of Figure 1 is missing
Lines 120-121
There is no data on the collection numbers of these strains
Line 124 “To protect the microbes in the artificial environment, the ant samples were ground only 2 weeks after collection.”
It is not clear what kind of protection the authors keep in mind.
In two weeks, the composition of ants microbiome could change significantly. And the data received after such a period may not reflect the natural state of things.
Line 125
It is not obvious whether 25 individuals were taken from each colony or only 25 individuals were examined
Line 125
Was each ant examined separately or were the mixed samples made?
Line 128
Also, “a liquid sample” is an awkward term.
Line 130
Please specify which “9 sets of sterile screening” media were used
Despite the earlier remark, the authors use the word “medium” when talking about the plural
Line 131
What is “actinone”?
Specify the final concentrations of antibiotics in the medium
Line 132
The authors did not use any methods to isolate and purify DNA for amplification, that raises doubts about the quality of the obtained results
Lines 136-138
The sentence is grammatically contradictory
Lines 157
It was not worth replacing 50 mL by “Fifty microliters»
What is meant by the term “….activated actinomycetes”?
Lines 164-165
“One milliliter of the activated broth was added to a 250 ml flask”
The values and their names are indicated by both words and symbols
Lines 169-170
“Various amounts of filtrate (1 ml, 1.5 ml, and 2 ml) were added to sterilized PDA medium (4 ml, 3.5 ml, and 3 ml) with a diameter of 5.5cm.”
What does the value of 5.5 cm refer to?
Line 172-173
“Plant pathogenic fungal cakes”
Sounds scary… The authors should not invent terms, but use those that are accepted in the literature!
Line 177, 190, 200, 201
“…symbiotic bacteria”
This remark has already been made before, there is no evidence of symbiosis
Figures 2, 3
The information for the 16S phylogeny reconstruction is not adequate. Although the software MEGA is widespread tool to construct phylogenetic tree, however, many options and criteria are available or could be applied in MEGA. Depending on the selected options, tree topology may differ. Therefore, a full disclosure allows reproducibility/repeatability of the tree topology.
Also, the neighbour-joining, maximum-likelihood and maximum parsimony trees are normally presented to check for congruence albeit recent trends support the primacy of maximum-likelihood method to infer phylogenetic relationships.
How many nucleotides/bp/sites (after alignment with/without trimming?) were used for the phylogeny tree? What was the treatment for handling gaps and missing data after alignment?
Lines 240-241
The assumption that strain B may belong to a new species has no apparent basis. The reasoning about the new species is based on the analysis of DNA similarity. The ability to assimilate substrates may vary between strains of the same species.
Figures 6-9
Such a number of photos is excessive
Lines 349-369
The authors promise to conduct various studies and obtain new data. I think it would be better to implement these plans, and then prepare the publication. The present work has no scientific value, since it is clearly not finished.
Lines 385 References
The list of references is still neglectful and not designed in accordance with the rules.
I strongly recommend reading “Instructions for Authors” https://www.mdpi.com/journal/insects/instructions#preparation
https://www.mdpi.com/authors/references
Author Response
We have made the response point by point in the attachment.

Reviewer 2 Report
I carefully read the resubmitted version of the manuscript and considering the numerous changes made by Authors, I think that the present version of the manuscript should be accepted. Due to the presence of the track revision it has been impossible to evaluate the presence of a proper pagination so that I think that the editorial office should evaluate if it is adequate.
Author Response
We sincerely thank the recognition of reviewer 2. the Track Change function is now used in order to help reviewers track our revision.